# *Mentha* spp. Essential Oils: A Potential Toxic Fumigant with Inhibition of Acetylcholinesterase Activity on *Reticulitermes dabieshanensis*

**DOI:** 10.3390/plants12234034

**Published:** 2023-11-30

**Authors:** Ziwei Wu, Chunzhe Jin, Yiyang Chen, Shimeng Yang, Xi Yang, Dayu Zhang, Yongjian Xie

**Affiliations:** The Key Laboratory for Quality Improvement of Agricultural Products of Zhejiang Province, College of Advanced Agricultural Sciences, Zhejiang A&F University, Hangzhou 311300, China

**Keywords:** *Mentha*, fumigation activity, *Reticulitermes dabieshanensis*, binominal mixture, detoxifying enzyme, acetylcholinesterase

## Abstract

In this study, we analyzed the components of *Mentha* spp. essential oils (EOs) and evaluated their major constituents and binary combinations against *Reticulitermes dabieshanensis*. We also determined the activities of esterases (ESTs), glutathione S-transferases (GSTs), and acetylcholinesterase activity (AChE) in treated insects. According to our findings, the most effective oils were those obtained from *M. citrata* (with the major constituent linalool constituting 45.1%), *M. piperita* (menthol, 49.1%), and *M. spicata* (carvone, 69.0%), with LC_50_ values of 0.176, 0.366, and 0.146 μL/L, respectively. The LC_50_ values were recorded for linalool (0.303 μL/L), followed by menthol (0.272 μL/L), and carvone (0.147 μL/L). The insecticidal potency increased with binary mixtures of major active constituents, with carvone strongly synergizing the toxicity of linalool and menthol against *R. dabieshanensis*. Compared to the control, except for *M. citrata* treated with no difference in α-NA or GST activity, the activities of ESTs and GST in other treatment groups were significantly increased. Additionally, our results found that *Mentha* spp. EOs and their major constituents inhibited the activity of AChE in vivo and in vitro. Finally, we performed a structure-based virtual screening of linalool, menthol, and carvone to identify that linalool had the greatest potential to bind to the active site of AChE. The present study suggests that *Mentha* spp. EOs could provide an additional approach for the management of termites over synthetic insecticides.

## 1. Introduction

Termites comprise a group of social wood-feeding cockroaches with over 3000 species [1]. These pests are a significant threat to agriculture and forestry in tropical and subtropical regions and can cause extensive economic and social harm [2]. *Reticulitermes flaviceps* is responsible for causing serious damage to agriculture and forestry in China [3]. However, traditional methods of chemically controlling termites often have negative impacts on the environment and human health. Therefore, plant essential oils (EOs) are gaining more attention as a safe and effective insecticide to control termites.

*Mentha*, an aromatic herb, is a species widely cultivated and used in the Labiatae family. It has approximately 19 varieties and 13 natural hybrids [4]. Most *Mentha* plants are annual and widely distributed across Europe, Asia, Africa, and America [5]. It can be used as herbal medicine to treat colds, flu, fever, headaches, and other diseases [6], as well as as a tea drink [7]. *Mentha* plants are rich in essential oils, and their primary chemical components include linalool acetate, menthol, and carvone [8,9]. Previous studies have reported that the essential oils and extracts from *Mentha* plants exhibit robust antioxidant and insecticidal properties [7]. For instance, *Mentha spicata* (EO) demonstrates strong fumigation efficacy against *Acanthoscelides obtectus* [10], *Helicoverpa armigera* [11], *Plutella xylostella* [12], *Culex quinquefasciatus*, *Aedes aegypti*, and *Anopheles stephensi* [13]. Furthermore, *M. viridis* EO exhibits potent oviposition inhibitory activity against female adults of *Tetranychus turkestani* [14] and *Acanthoscelides obtectus* [10]. *M. piperita* has been found to have preferable contact activity against *T. urticae* [15], as well as *Sitophilus oryzae*, *Tribolium castaneum*, and *Rhizopertha dominica* [16,17]. *M. piperita* has also been found to affect the parasitism of *Amyloodinium ocellatum* and *Piaractus brachypomus* [18] and to have a certain oviposition inhibition effect on *Acanthoscelides obtectus* [19]. Additionally, the essential oil of *M. citrata* has a good contact-killing effect on *Lipaphis erysimi* [20]. However, few reports exist regarding the research of *Mentha* EO in controlling termites. Therefore, this study aimed to examine the biological activity and mechanism of *Mentha* EO on termites to explore additional plant resources for termite control and provide a scientific foundation for the development of safe and green new termite control agents.

The lipophilicity of EOs can interfere with the basic metabolism, physiology, biochemistry, and behavioral selection of insects. Additionally, certain plant EOs can affect the development, reproduction, and survival of insects, as supported by research conducted by Suwansirisilp et al. [21], Kulkarni et al. [22], and Kumrungsee et al. [23]. A study found that the essential oil of green mint can inhibit the activity of detoxification enzymes in termites, resulting in termite death. Furthermore, its main components also display potent insecticidal activity against termites, as demonstrated by Yang et al. [3].

The objective of this study was to assess the insecticidal properties of three *Mentha* EOs and their major constituents and mixed compounds against *Reticulitermes dabieshanensis* and to determine the mechanism of toxicity through in vivo and in vitro enzyme activity determination and molecular docking simulation in order to establish a theoretical foundation for the development of new termite control agents. Additionally, chromatography-mass spectrometry (GC–MS) was used to analyze the components of these essential oils.

## 2. Results

### 2.1. Chemical Composition of Mentha spp. EOs

The chemical composition of *Mentha* spp. EOs was determined by GC–MS. The percentage fractions of each constituent in the total composition are listed in Table 1 and Appendix A. In *M. citrata* EO, a total of 10 constituents were found, accounting for 97.5% of the EO mass. The primary constituent was linalool (45.1%), while other constituents present at a noteworthy level were linalyl acetate (42.9%), caryophyllene (2.5%), and α-terpineol (2.1%). *M. piperita* EO contained 10 compounds, constituting 99.9%, and menthol (49.1%), menthone (27.5%), and isomenthol (5.6%) were the major constituents. *M. spicata* EO had a total of 5 identified constituents, accounting for 98.1%. The major constituent was carvone (69.0%), and other significant constituents were limonene (17.9%), menthol (8.5%), and menthone (2.2%).

### 2.2. Vapor Activity of Mentha spp. EO and the Major Constituent

The results achieved revealed that three essential oils and major constituents had dose-dependent insecticidal effects against *R. dabieshanensis* (Figure 1) compared to the control, which induced no mortality at the concentration tested after 24 h of exposure. According to Table 2, the LC_50_ values of *M. citrata*, *M. piperita*, *M. spicata* EOs, linalool, menthol, and carvone against *R. dabieshanensis* were 0.176, 0.366, 0.146, 0.303, 0.272, and 0.147 μL/L, respectively.

### 2.3. Fumigation Activity of Binary Mixtures

The binary mixtures of carvone (LC_30_) + linalool (LC_30_), carvone (LC_30_) + linalool (LC_50_), carvone (LC_30_) + menthol (LC_30_), and carvone (LC_30_) + menthol (LC_30_) in all the ratios showed a synergistic effect against *R. dabieshanensis* (Table 3).

### 2.4. ESTs, GST, and AChE Enzyme Activities

The *R. dabieshanensis* treated by *M. piperita* and *M. spicata* EOs, along with linalool, menthol, and carvone, demonstrated increased activity of esterases (for α-NA, F = 18.080, d.f. = 6, 14, *p* = 0.0001; for β-NA, F = 89.120, d.f. = 6, 14, *p* < 0.0001). However, the highest esterase activity was recorded in the larvae treated with *M. spicata* (α-NA) and *M. citrata* (β-NA) (Figure 2). The activity of GSTs significantly increased in *R. dabieshanensis* exposed to *M. piperita* and *M. spicata* EOs, linalool, menthol, and carvone in comparison to the control (F = 29.821, d.f. = 6, 14, *p* < 0.0001) (Figure 2). However, there were no differences in the activity of α-NA or GSTs between the *M. citrata* and control groups. Conversely, the activity of acetylcholinesterase significantly decreased in all treatments (F = 19.872, d.f. = 6, 14, *p* < 0.0001), with *M. piperita* EO exhibiting the highest inhibition activity among the oils and compounds tested.

The in vitro inhibitory AChE effects of *Mentha* spp. EO against *R. dabieshanensis* increased significantly as the concentration increased (for *M. citrata* EO, F = 16.806; df = 4, 10; *p* < 0.001; for *M. piperita* EO, F = 36.839; df = 4, 10; *p* < 0.001; for *M. spicata* EO, F = 46.694; df = 4, 10; *p* < 0.001; Figure 3A–C). According to Table 4, the IC_50_ values of *M. citrata*, *M. piperita*, and *M. spicata* EOs against *R. dabieshanensis* were 18.295, 0.765, and 3.228 μL/mL, respectively. Additionally, each tested compound (linalool and menthol) also showed a significant variation in AChE inhibitory activity at different concentrations (for linalool, F = 28.551; df = 4, 10; *p* < 0.001; for menthol, F = 32.160; df = 4, 10; *p* < 0.001; Figure 3D,E), but carvone has no significant difference (F = 3.433; df = 4, 10; *p* = 0.052; Figure 3F), with an IC_50_ of 0.136, 76.790, and 1.922 μL/mL, respectively (Table 4).

### 2.5. Molecular Docking

Molecular docking of compounds linalool, menthol, and carvone with the acetylcholinesterase protein model (*R. dabieshanensis*) was carried out using the Discovery Studio 2019 software packages. The molecular docking results of linalool, menthol, and carvone are shown in Figure 4. From the results, compounds containing a hydroxyl group exhibited similar binding modes with acetylcholinesterase (Figure 4A), and the hydroxyl group played a vital role in binding, which formed a hydrogen bond and an attractive change interaction with His727B and Glu486B. For compound menthol (Figure 4B), with its great fumigant toxicity against *R. dabieshanensis*, the hydroxyl group interacts with Glu575B and forms a hydrogen bond. The H atom and O atom formed alkyl interactions with Glu575B, respectively. For compound carvone (Figure 4C), which has the best fumigant toxicity against *R. dabieshanensis*, the carbonyl group of carvone interacts with the hydroxyl group of Ser487B to form hydrogen bonds. These docking results indicated that compounds like linalool, menthol, and carvone could readily dock into the different binding sites of acetylcholinesterase. The docking results further revealed that the three compounds could be promising acetylcholinesterase inhibitors.

## 3. Discussion

Essential oils are characterized by being easily volatile, having low residuality, and having multiple modes of action, making them often considered substitutes for chemical pesticides. They can effectively control pests in small-scale agricultural systems [2,24]. *Menthas* are famous spices widely used in food, beverages, pharmaceuticals, and cosmetics for their valuable EOs. With their low cost, low environmental impact, and high safety for humans, mint plants are widely cultivated in Europe, Asia, Africa, and the Americas. *Mentha* EOs can be utilized in sustainable pest control.

Our results demonstrate that *M. citrata*, *M. piperita*, and *M. spicata* EOs had strong vapor activity in *R. dabieshanensis*. In a similar investigation carried out by Kumar et al. [25], they reported that *M. piperita* and *M. citrata* essential oils showed high toxicity against *Musca domestica.* L. Kedia et al. [26] demonstrated the insecticidal efficacy of *M. spicata* EO, showing 98.46% oviposition deterrency, 100% ovicidal activity, 88.84% larvicidal activity, 72.91% pupaecidal activity, and 100% antifeedant activity against *Callosobruchus chinensis* at 0.1 μL/mL. Govindarajan et al. [13] showed *M. spicata* EO to have toxic effects on *C. quinquefasciatus* (LC_50_ = 62.62 ppm), *Aedes aegypti* (LC_50_ = 56.08 ppm) and *Anopheles stephensi* (LC_50_ = 49.71 ppm). Studies also reported the effectiveness of *M. piperita* EO against *Liposcelis bostrychophila* [27], *Culex pipiens* [28], *Tetranychus urticae* [29], *Phthorimaea absoluta* [30], *Tribolium castaneum*, and *Sitophilus oryzae* [16,17]. The results of the present study and the previous ones by several authors showed similar trends in the toxicity of the essential oils used, even in different orders of insects.

Here, *M. citrata* and *M. spicata* EOs demonstrated the highest insecticidal activity in comparison to their constituents against *R. dabieshanensis*. The latest report shows that the main component, eugenol, of *Ocimum basilicum* EO has the highest insecticidal activity against *R. dabieshanensis* [31]. Similarly, Piri et al. [32] found that Ajwain EO exhibited the highest insecticidal activity against *Tuta absoluta* larvae compared to its components. Shahriari et al. [33] found that the Ajwain EO was more toxic to *Ephestia kuehniella* larvae than thymol. In the present study, *M. piperita* EO was less toxic to *R. dabieshanensis* than their main constituent, menthol, as has also been reported in some previous studies [12,34]. Strong toxicity of linalool (LC_50_ = 2.11 μL/L) and carvone (10.87 μL/L) against *Callosobrunchus maculatus* has been reported by Oyedeji et al. [35]. Hussein et al. [36] showed high toxicity against *Aphis nerii*. The observed insecticidal activity of *Mentha* EOs in the current study could be related to the synergistic/antagonist effects of the individual components within the EO. Previous studies have shown that a mixture of several active compounds in essential oils enhances their respective insecticidal activity [3,32].

The mixture of EOs/extracts increases the spectrum of action of insecticides, as different species have different reactions to individual EOs/extracts. The study by Kim et al. [37] demonstrated that the binary mixtures of basil and mandarin EOs against *Spodoptera litura* resulted in a synergistic interaction. According to Aungtikun et al. [38], combinations of *Cymbopogon citratus*, *Illicium verum*, and *Myristica fragrans* resulted in synergistic effects on *Musca domestica*, respectively. By contrast, Haouel-Hamdi et al. [39] demonstrated that the mixtures of *Mentha rotundifolia* and *M. longifolia* EOs showed antagonistic effects on *Sitophilus oryzae*. Additionally, binart mixtures of *Moringa olerifera*, *Azadirachta indica*, and *Eucalyptus globulus* leaf extract were synergistically effective against *Diurophous noxia* [40]. The major constituents of EOs are blended to form a new formulation (binary mixtures), which has additive or synergistic properties upon toxicity. In our study, carvone strongly synergized the toxicity of linalool and menthol against *R. dabieshanensis*. Similarly, Pavela [41,42] demonstrated that the binary mixtures of carvone and linalool resulted in synergistic effects on the adult mortality of *Spodoptera littoralis* and *C. quinquefasciatus*, respectively. Additionally, ocimene strongly synergized the toxicity of β-myrcene, L-limonene, geraniol, and L-menthol against *Planococcus lilacinus* [43]. In addition, the study of Prasannakumar et al. [30] revealed that the combined use of 4-carene and α-pinene was very effective against *Phthorimaea absoluta*. In terms of individual compounds, the binary mixture activity of pure compounds may be more pronounced because they have many mechanisms of action and may delay the emergence of resistance in pests.

Furthermore, earlier studies by Pavela et al. [44,45] and Sánchez-Gómez et al. [46], indicating exposure to effective insecticidal essential oils LC_30_ or LD_30_ has an impact on the development of *C. quinquefasciatus* and *S. littoralis* larvae, as well as on the longevity of adult *Musca domestica*, have been reported. Considering that there is little information on the physiological effects of essential oils at low lethal concentrations/doses (i.e., LC_30_ and LD_30_) on termites. Therefore, the impact of exposure to effective insecticidal essential oils and their main components, LC_30_ or LD_30_, on the EST and GST detoxifying enzymes of termites was evaluated.

ESTs and GST detoxifying enzymes are involved in the metabolism of botanical insecticides and are typically enhanced by exogenous compounds [31]. The present study found that exposing *R. dabieshanensis* to LC_30_ of *Mentha* spp. EOs and their major constituents significantly increased the levels of ESTs and GSTs in vivo. Our study is similar to earlier studies by Shahriari et al. [33], Piri et al. [32], Wang et al. [47], Kumrungsee et al. [22], Wei et al. [48], and Yang et al. [3], indicating enhancements in detoxifying enzymes from insect treatments with EOs and their compounds. Earlier studies suggested the toxicity of EOs constituents may be related to cell damage in insect tissues [48,49,50]. Given the importance of these enzymes in cell protection from plant allelochemicals, the induction of their activity can be considered a defense mechanism in *R. dabieshanensis* against EO. 

AChE plays an important role in the mechanism of action of certain EOs or their constituents to produce insecticidal effects [45]. The present study demonstrates that EO and its major compounds significantly inhibit acetylcholinesterase of *R. dabieshanensis* in vivo and in vitro, which has also been reported by Piri et al. [32], Wang et al. [44], and Yang et al. [31]. In this study, we performed structure-based virtual screening of linalool, menthol, and carvone to identify compounds with the potential to produce binding profiles. Our results indicate that linalool has the greatest potential to bind to the active site of AChE. This finding is consistent with that of [51], where docking simulations showed that linalool (approximately −40 kcal/mol) exhibited the most favorable energy grid scores when compared to the other 24 compounds.

## 4. Materials and Methods

### 4.1. Insects 

*R. dabieshanensis* was collected at Linglong Mountain (30.2251°N, 119.6843°E) in Lin’an District, Hangzhou. They were fed with water and newspaper and were maintained in an incubator (27 ± 1 °C, 80 ± 5% relative humidity, L:D = 0:24 h) in the laboratory. We selected healthy and uniform-sized workers for the experiments. 

### 4.2. Mentha spp. EOs and the Constituents

*Mentha citrata*, *M. piperita*, and *M. spicata* EOs were obtained from Poli Co., Ltd. (Shanghai, China); The major constituents, carvone (97%), menthol (98%), and linalool (95%), were purchased from Sigma-Aldrich (Shanghai, China) and stored at 4 °C until the experiments.

### 4.3. GC-MS Analysis

The chemical composition of *Mentha* spp. EOs was analyzed using GC-MS. The capillary pillar used was HP-5MS with a dimension of 30 m × 0.25 mm i.d. and 0.25 μm film thickness, and a sample volume of 1.0 μL was injected. The carrier gas flow rate of helium was set at 1.0 mL/min, with a split ratio of 1:50 and an initial column temperature of 50 °C for 10 min. The temperature was raised, then increased at a rate of 10 °C/min until it reached 250 °C. The inlet temperature was set at 250 °C. Mass spectrometry conditions were set using an EI ion source with a voltage of 70 eV, an ion source temperature of 220 °C, an interface temperature of 250 °C, and a scanning range of 15–500 m/z. Chemical compositions were identified using the NIST11.LIB database and literature [52] by comparing their retention index.

### 4.4. Fumigant Toxicity 

The fumigation activity of *R. dabieshanensis* was tested using the method described by Yang et al. [3]. A qualitative filter paper measuring 1.5 × 6 cm was affixed to the lid of a glass tank with a capacity of 1 L (10 cm diameter × 12.5 cm), and then three kinds of EOs and three main components were added to the filter paper, respectively. The acetone treatment was used as the control. Five concentrations of each substance were selected, and each concentration had three replicates, and 20 workers with similar body vitality were selected for each replicate. The experiment was conducted in an insect incubator set at 25 ± 1 °C and 75 ± 5% RH under dark conditions. After 24 h, the termites were observed, and the number of deaths was recorded.

### 4.5. Fumigation Activity of Binary Mixtures

The binary mixtures of the major constituents of *Mentha* EO (carvone, menthol, and linalool) were performed using the method of Yang et al. [3]. The three compounds were combined in a 1:1 ratio at doses of LC_30_:LC_30_, LC_30_:LC_50_ and LC_50_:LC_30_. Each dose was tested on 20 workers three times.

Observed mortalities were compared to expected mortalities using the formula:E = Oa + Ob(1 − Oa)
where E is expected mortality, Oa and Ob are the observed mortality of the first and second compounds used in the binary mixtures, respectively.

The effects of binary mixtures were determined as antagonistic, additive, or synergistic using χ^2^ comparative analysis.
χ^2^ = (Om − E)^2^/E
where E is the expected mortality from the binary mixture and Om is observed mortality, when χ^2^ > 3.84 and χ^2^ < 3.84 were considered synergistic or additive effects [32,42].

### 4.6. Determination of Enzyme Activity

#### 4.6.1. Enzyme Assays

The worker adults of termites were exposed to the LC_30_ concentrations of *Mentha* spp. EOs and their major constituents to determine their effects on the esterases, glutathione S-transferases, and acetylcholine esterase. Samples were prepared by homogenizing 10 termites in 1 mL phosphate buffer (0.1 M, pH 7.0), centrifuging 12,000× *g* for 15 min at 4 °C, and storing the supernatants at −80 °C for late use. All biochemical tests were repeated three times.

#### 4.6.2. Esterases (ESTs)

The ESTs test was determined by using the method of Piri et al. [32]. To a 96-well microplate, 20 μL of 10 mM α-NA and β-NA and 50 μL of 1 mM Fast Blue RR Salt were added. Next, 10 μL of enzyme solution was added and incubated at 27 °C for 5 min. The absorbance (OD value) was then measured at 450 nm and recorded every 1 min for 10 min. A control group was set up with 10 μL of enzyme solution added to 10 μL of distilled water. Each experiment was repeated more than three times.

#### 4.6.3. Glutathione S-Transferases (GSTs)

GSTs activity was determined using the method described by Piri et al. [32]. In 96-well microtiter plates, 20 μL of 20 mM CDNB and 510 μL of enzyme solution were added. The mixture was incubated at 27 °C for 5 min, the OD value at 340 nm was measured, and the measurement was recorded every 1 min for a total of 10 min. Each experiment was repeated more than three times.

#### 4.6.4. In Vivo Acetylcholinesterase (AChE) Activity Assay

The activity of acetylcholinesterase was determined according to the method of Ellman et al. [53]. The reaction solution was incubated at 25°C for 5 min and consisted of 80 μL of 0.1 M phosphate buffer with a pH of 7.0, 50 μL 10 mM acetylcholine iodide, and 50 μL of 10 mM 5,5-dithiobis-2-nitrobenzoic acid (DTNB). The OD value was measured at 405 nm using a 96-well microplate reader.

#### 4.6.5. In Vitro Acetylcholinesterase Activity Assay

In an in vitro AChE activity assay, ten termiters were ground using a porcelain mortar in 0.1 M Tris-HCl buffer (pH 7.8), which contained NaCl (20 mM) and 0.5% Triton X-100. The resulting mixture was then centrifuged at 15,000× *g* for 15 min at 4 °C. *M. piperita* EO and menthol (diluted to 1.0, 5.0, 10.0, 15.0, and 20.0 μL/mL in 99.5% alcohol), as well as *M. citrata*, *M. spicata* EOs, linalool, and carvone (0.5, 1.0, 2.0, 4.0, and 6.0 μL/mL, *v*/*v* in alcohol), were prepared. Afterwards, 20 μL of each constituent was mixed separately with 40 μL of enzyme solution, 50 μL acetylthiocholine iodide (10 mM), 10 μL DTNB (4 mM), and 100 μL of Tris-HCl buffer in a 96-well microplate. The mixture was incubated for 30 min before being read at absorbance at 405 nm, with alcohol being used as a control. 

### 4.7. Homology Modeling and Molecular Docking

The Uniprot database (https://www.uniprot.org/, accessed on 10 September 2023) was used to search the amino acid sequence of the acetylcholinesterase from *R. dabieshanensis*. The protein sequence A0A6L2PI50_COPFO was modeled using the acetylcholinesterase template from *Coptotermes formosanus*, and the 3D structure was established with SWISS-MODEL (http://swissmodel.expasy.org/, accessed on 10 September 2023). The model was evaluated using Ramachandran. The energy-minimizing structures of linalool, menthol, and carvone were obtained with Discovery Studio 2019. The molecular docking of these compounds with acetylcholinesterase was performed by Auto Dock Tools. The bling box for Auto Dock Tools was defined as x, y, and z with specified coordinates (e.g., x = −1.911, y = −17.384, z = 24.253), and other parameters were set to their default values. Subsequently, the optimal pose with the best docking score was selected and visualized.

### 4.8. Statistical Analysis

The results were expressed as means ± standard errors. The doses causing 50% and 90% (LC_50_ and LC_90,_ respectively) mortality were determined by probit analysis using the Online Tool (OPSTAT) (http://14.139.232.166/Probit/probitanalysis.html, accessed on 30 October 2023). The inhibition rate was tested through the analysis of variance (ANOVA), followed by Duncan’s new multiple range method Significant Difference test at the *p* = 0.05 level of significance using SPSS (version 19.0; SPSS Inc., Chicago, IL, USA).

## 5. Conclusions

Our results indicate that *M. citrata*, *M. piperita*, and *M. spicata* EOs and their major compounds, linalool, menthol, and carvone, are highly toxic to termites. This toxicity is achieved through the inhibition of AChE activity, which suggests that these compounds could be developed as control agents for termites. However, before future application in the field, it is necessary to determine the effects of these active substances on non-target organisms, as well as to design sustained-release formulations to improve the durability of essential oils. This may lead to effective management of underground termites.

## Figures and Tables

**Figure 1 plants-12-04034-f001:**
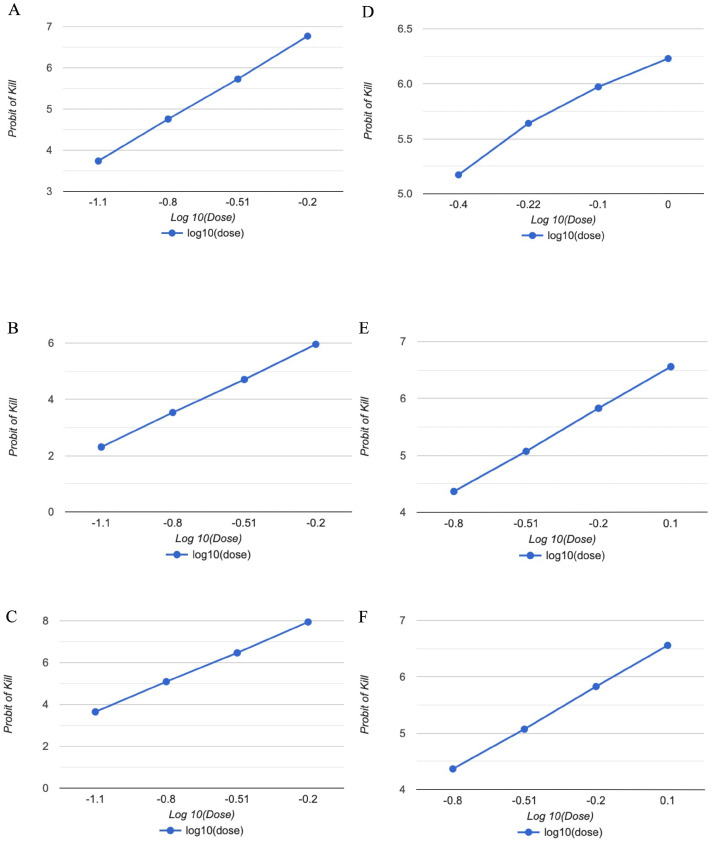
Variation in *R. dabieshanensis* mortality as a function of essential oils [(**A**): *M. citrata*; (**B**): *M. piperita*; (**C**): *M. spicata*; (**D**): Linalool; (**E**): Menthol; (**F**): Carvone] doses by contact as per the probit transformation model.

**Figure 2 plants-12-04034-f002:**
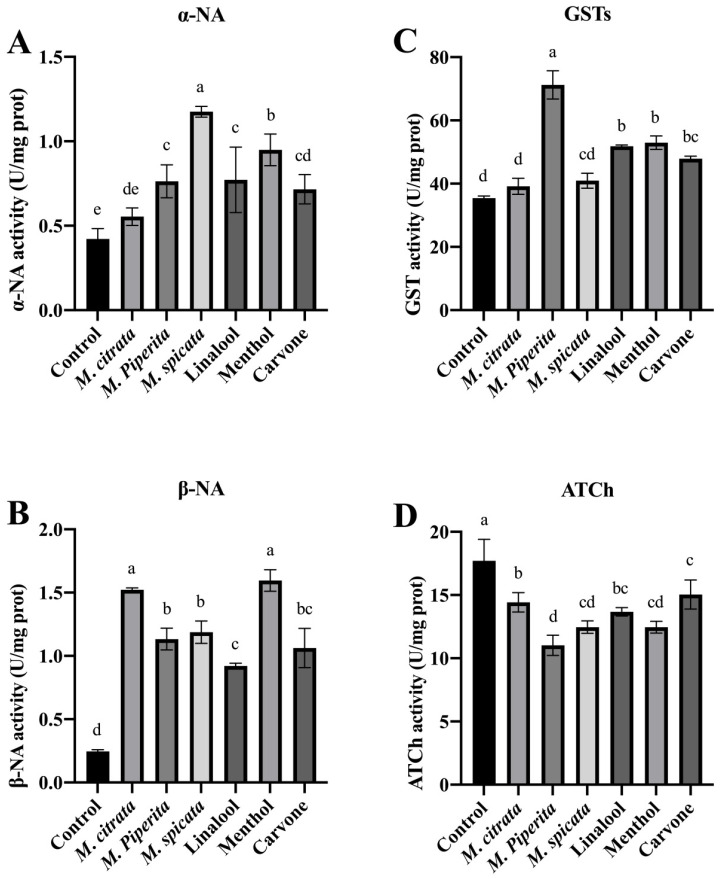
The effects of *Mentha* spp. EOs and their major constituents on the activities of esterases, glutathione S-transferase, and acethylcholine esterase (U/mg protein, respectively), of *R. dabieshanensis*. (**A**): activity of esterases (α-NA) for 24 h of LC_30_ treatment; (**B**): activity of esterases (β-NA) for 24 h of LC_30_ treatment; (**C**): activity of GSTs for 24 h of LC_30_ treatment; (**D**): activity of AChE for 24 h of LC_30_ treatment. Means (±SD) values with different letters (a–e) are significantly different at a level of *p* < 0.05, according to Duncan’s test.

**Figure 3 plants-12-04034-f003:**
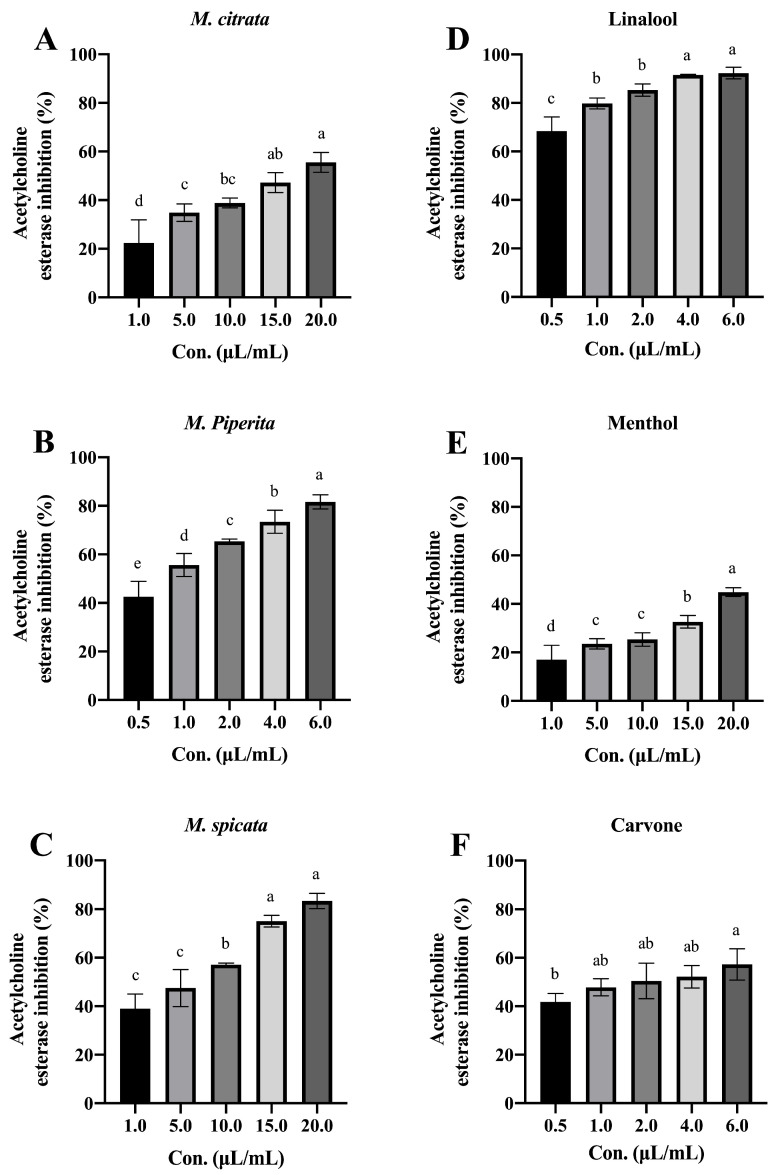
In vitro assay for estimation of acetylcholine esterase inhibition activity of *Mentha* spp. EOs and their major constituents against *R. dabieshanensis*. (**A**): *M. citrata*; (**B**): *M. piperita*; (**C**): *M. spicata*; (**D**): Linalool; (**E**). Menthol; (**F**). Carvone. Means (±SD) values with different letters (a–e) are significantly different at a level of *p* < 0.05, according to Duncan’s test.

**Figure 4 plants-12-04034-f004:**
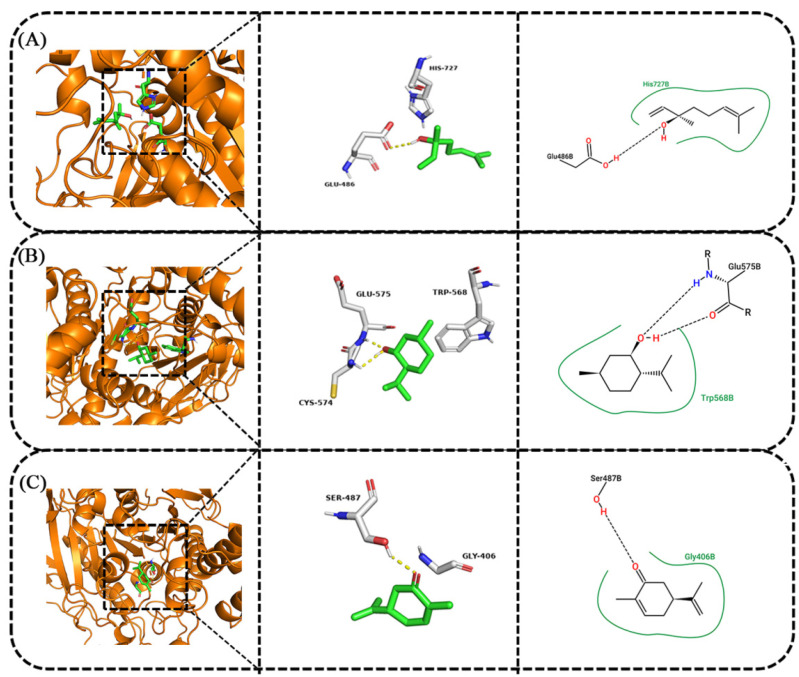
Molecular docking diagrams of compounds. (**A**): Linalool; (**B**): Menthol; (**C**): Carvone.

**Table 1 plants-12-04034-t001:** Major chemical constituents of *Mentha* spp. essential oils.

No.	Constituents	RI_Exp._	RI_Lit._	Relative Area (%)		
				*M. citrata*	*M. piperita*	*M. spicata*
**1**	α-Pinene	939	932	-	0.7	-
**2**	β-Pinene	979	979	0.9	0.9	-
**3**	β-Myrcene	991	990	0.6	-	-
**4**	Carane	1010	1009	-	5.5	0.5
**5**	Limonene	1027	1027	1.6	5.2	17.9
**6**	Linalool	1097	1098	45.1	-	-
**7**	Isopulegol	1141	1145	-	0.7	-
**8**	Menthone	1149	1151	-	27.5	2.2
**9**	Isomenthol	1155	1158	-	5.6	-
**10**	Menthol	1170	1167	-	49.1	8.5
**11**	α-Terpineol	1191	1191	2.1	-	-
**12**	Pulegone	1235	1235	-	1.5	-
**13**	Carvone	1243	1240	-	-	69.0
**14**	Linalyl acetate	1253	1252	42.9	-	-
**15**	Neryl acetate	1356	1358	0.5	-	-
**16**	Geranyl acetate	1380	1379	0.8	-	-
**17**	Caryophyllene	1419	1417	2.5	3.2	-
**18**	Caryophyllene oxide	1587	1589	0.5	-	-
	Total			97.5	99.9	98.1

**Table 2 plants-12-04034-t002:** LC_50_ values (μL/L) of *Mentha* spp. EOs and their major constituents against *R. dabieshanensis*.

EOs	LC_30_ (95% CI)	LC_50_ ^a^ (95% CI ^b^)	LC_90_ (95% CI)	Slope ± SD	(χ^2^) ^c^	*p* Value
** *M. citrata* **	0.119 (0.088–0.149)	0.176 (0.139–0.225)	0.460 (0.336–0.770)	0.25 ± 0.2867	29.156	0.006
** *M. piperita* **	0.272 (0.219–0.325)	0.366 (0.306–0.449)	0.756 (0.585–1.181)	0.205 ± 0.3583	22.459	0.049
** *M. spicata* **	0.116 (0.089–0.139)	0.146 (0.119–0.178)	0.256 (0.205–0.385)	0.2883 ± 0.3416	34.256	0.001
**Linalool**	0.176 (0.114–0.230)	0.303 (0.233–0.362)	1.130 (0.889–1.674)	0.1317 ± 0.285	9.893	0.703
**Menthol**	0.164 (0.132–0.196)	0.272 (0.230–0.320)	0.936 (0.731–1.318)	0.2183 ± 0.1183	10.952	0.615
**Carvone**	0.111 (0.083–0.133)	0.147 (0.121–0.178)	0.290 (0.226–0.483)	0.2367 ± 0.2533	42.123	0.001

^a^ LC_50_, or Median lethal concentration, is the concentration at which half of the insects died. ^b^ CI_95_, Confidence intervals the activity of two compounds was considered significantly different when the 95% CIs failed to overlap. ^c^ Pearson’s chi-square, suggesting the goodness of fit test, is not significant (ns) at *p* > 0.05.

**Table 3 plants-12-04034-t003:** Relative toxicity of binominal mixtures of the major constituents of *Mentha* spp. *EOs* against *R. dabieshanensis* after 24 h of post-treatment.

Binary Mixtures	Con.	N	Mortality (%)				χ^2^	Effect
			Pure Compound		Binary Mixtures			
			Oa	Ob	Em	Om		
Carvone + Linalool	LC_30_ + LC_30_	60	35.00	6.67	39.33	53.33	4.99	Synergy
Carvone + Linalool	LC_30_ + LC_50_	60	35.00	25.00	51.25	41.67	1.79	Addtive
Carvone + Linalool	LC_50_ + LC_30_	60	51.67	6.67	54.89	83.33	14.74	Synergy
Carvone + Menthol	LC_30_ + LC_30_	60	35.00	25.00	51.25	33.33	6.26	Synergy
Carvone + Menthol	LC_30_ + LC_50_	60	35.00	45.00	64.25	51.67	2.46	Addtive
Carvone + Menthol	LC_50_ + LC_30_	60	51.67	25.00	63.75	81.67	5.04	Synergy
Linalool + Menthol	LC_30_ + LC_30_	60	6.67	25.00	30.00	33.33	0.37	Addtive
Linalool + Menthol	LC_30_ + LC_50_	60	6.67	45.00	48.67	48.33	0.00	Antagonist
Linalool + Menthol	LC_50_ + LC_30_	60	25.00	25.00	43.75	33.33	2.48	Addtive

**Table 4 plants-12-04034-t004:** In vitro assay for estimation of inhibition concentration rate (IC_50_, μL/mL) for *Mentha* spp. EOs and their major constituents.

EOs	IC_50_ ^a^ (95% CI ^b^)	Slope ± SD	(χ^2^) ^c^	*p* Value
** *M. citrata* **	18.295 (12.681–32.359)	0.0787 ± 0.1615	20.597	0.810
** *M. piperita* **	0.765 (0.596–0.931)	0.0961 ± 0.3489	9.795	0.711
** *M. spicata* **	3.228 (1.780–4.782)	0.1163 ± 0.2551	48.124	0.001
**Linalool**	0.136 (0.066–0.218)	0.0596 ± 0.656	7.738	0.860
**Menthol**	76.790 (36.58–166.367)	0.0647 ± 0.093	21.700	0.060
**Carvone**	1.922 (1.131–3.308)	0.0353 ± 0.3933	12.262	0.506

^a^ LC_50_, or Median lethal concentration, is the concentration at which half of the insects died. ^b^ CI_95_, Confidence intervals the activity of two compounds was considered significantly different when the 95% CIs failed to overlap. ^c^ Pearson’s chi-square, suggesting the goodness of fit test, is not significant (ns) at *p* > 0.05.

## Data Availability

The data set used and analyzed during the present study are available from the corresponding author upon reasonable request.

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
