# Peer review of "Mentha spp. Essential Oils: A Potential Toxic Fumigant with Inhibition of Acetylcholinesterase Activity on Reticulitermes dabieshanensis"

_plants, 2023, doi:10.3390/plants12234034_

Round 1

Reviewer 1 Report

Comments and Suggestions for Authors

In this study, the authors analyzed the components of Mentha spp. essential oils (EOs) and evaluated their major constituents and binary combinations against Reticulitermes dabieshanensis. It is an interesting article that adds to the knowledge about the mechanism of effects of EOs on arthropods. The article can be published after the following minor modifications.

1. The authors found the direct effectiveness of EOs on Reticulitermes dabieshanensis, but it would be appropriate to discuss also the effectiveness of lethal (sublethal) doses in the practical context use of EOs in insect protection. EOs can reduce the vitality, fertility of insects and affect behavior (see e.g. 10.1016/j.indcrop.2021.113590; 10.1016/j.indcrop.2021.114413; DOI: 10.1016/j.fct.2019.111037 ). How applications of e.g. LC30 might translate to termites: Discuss.

2. Given the pesticidal effects of EOs, briefly discuss the environmental and health safety of potential botanical insecticides based on mint EOs.

3. The conclusion should be more practically focused, i.e. how the knowledge gained can be used in practice.

Author Response

In this study, the authors analyzed the components of Mentha spp. essential oils (EOs) and evaluated their major constituents and binary combinations against Reticulitermes dabieshanensis. It is an interesting article that adds to the knowledge about the mechanism of effects of EOs on arthropods. The article can be published after the following minor modifications.

  1. The authors found the direct effectiveness of EOs on Reticulitermes dabieshanensis, but it would be appropriate to discuss also the effectiveness of lethal (sublethal) doses in the practical context use of EOs in insect protection. EOs can reduce the vitality, fertility of insects and affect behavior (see e.g. 10.1016/j.indcrop.2021.113590; 10.1016/j.indcrop.2021.114413; DOI: 10.1016/j.fct.2019.111037). How applications of e.g. LC30might translate to termites: Discuss.

Response: Thank you very much for your valuable advice. In discussion, we added “Furthermore, the earlier studies by Pavela et al. [44, 45] and Sánchez-Gómez et al. [46], indicating exposure to effective insecticidal essential oils LC30 or LD30 has an impact on the development of Culex quinquefasciatus and Spodoptera littoralis larvae, as well as on the longevity of adult Musca domestica have been reported. Considering that there is little information on the physiological effects of essential oils at low lethal concentrations/doses (i.e. LC30 and LD30) on termites. Therefore, the impact of exposure to effective insecticidal essential oils and their main components, LC30 or LD30, on the EST and GST detoxifying enzymes of termites was evaluated.”

  1. Given the pesticidal effects of EOs, briefly discuss the environmental and health safety of potential botanical insecticides based on mint EOs.

Response: Thank you for your valuable comments. In discussion, we added “Mentha are famous spices widely used in food, beverages, pharmaceutics, and cosmetics for their valuable EOs. With its low cost, low environmental impact, and high safety for humans, mint plants are widely cultivated in Europe, Asia, Africa, and the Americas countries. Mentha EOs can be utilized in sustainable pest control”. 

  1. The conclusion should be more practically focused, i.e. how the knowledge gained can be used in practice.

Response: Thank you for your valuable comments. In Conclusions, “improve the efficacy and durability through formulation modification, or use as a synergist for other insecticides, which could be effective in managing resistance.” was corrected as “designing sustained-release formulations to improve the durability of essential oils, this may achieve effective management of underground termites”.

Reviewer 2 Report

Comments and Suggestions for Authors

My main concern about this study is GC MS analysis. Essential oils are generally complex mixtures of compounds. And in this study are only 10 or less compounds detected which lead me to a conclusion that applied method is not appropriate. Please add chromatograms for all three samples (can be supplementary material) to clarify this question.

Linalool is considered as main compound in M.Citrata essential oil, but second compound Linalyl acetate is not considered. Since in table relative percentages are presented and quantification (using internal or external standard) is not conducted, can we really claim that linalool is responsible for mentha citrata bioactivities?

On the other hand, in mentha spicata, carvone percentage content was almost 70% and when you compare LC50 values were almost the same for pure compound and for essential oil.er Shouldn’t carvone have much higher bioactivity when it is present in a higher concentration if it is major compound 8as you stated)?

Line 183 please add reference for rapid degradation 

Line 256-259. Is this common procedure? “strong workers” sound a bit non usual. Please add reference.

Line 261-269. Please add all used chemicals. Include purity for used standards (carvone, menthol and linalool)

Line 331 please indicate ethanol concentration

Line 331 “M. citrata, M. spicata” should be written in italic, please check whole manuscript

Line 332, line 336 and several more have double space please check

Above all this paper has high similarity index (around 11%) with paper https://www.sciencedirect.com/science/article/abs/pii/S0926669021006580?via%3Dihub

Which is not acceptable for high impact factor journal (or any other).

Author Response

My main concern about this study is GC MS analysis. Essential oils are generally complex mixtures of compounds. And in this study are only 10 or less compounds detected which lead me to a conclusion that applied method is not appropriate. Please add chromatograms for all three samples (can be supplementary material) to clarify this question.

Response: Thank you for your valuable comments.We have added the chromatograms of three samples as supplementary materials.

Linalool is considered as main compound in M. citrata essential oil, but second compound Linalyl acetate is not considered. Since in table relative percentages are presented and quantification (using internal or external standard) is not conducted, can we really claim that linalool is responsible for mentha citrata bioactivities?

Response: Thank you very much for your valuable advice. Our previous study found that Linalyl acetate LC50 is 0.712 μL/L against R. dabieshanensis, which is twice as high as linalool (0.303 μL/L) (10.3390/molecules28052007).

On the other hand, in mentha spicata, carvone percentage content was almost 70% and when you compare LC50 values were almost the same for pure compound and for essential oil.er Shouldn’t carvone have much higher bioactivity when it is present in a higher concentration if it is major compound 8as you stated)?

Response: Thank you for your valuable comments. We believe that because the percentage content of carvone is almost 70%, when comparing the LC50 values of pure compounds and essential oils, they are almost the same.

Line 183 please add reference for rapid degradation

Response: Thank you very much for your valuable advice. There is a discussion on the rapid degradation of plant essential oils in the 24th reference.

Line 256-259. Is this common procedure? “strong workers” sound a bit non usual. Please add reference.

Response: Thank you for your valuable comments. We have already “strong” was corrected as “uniform size .

Line 261-269. Please add all used chemicals. Include purity for used standards (carvone, menthol and linalool)

Response: Thank you very much for your valuable advice. the major constituents, carvone (97%), menthol (98%), and linalool (95%), were purchased from Sigma-Aldrich (Shanghai, China).

Line 331 please indicate ethanol concentration

Response: Thank you for your valuable comments. We used 99.5% ethanol to dissolve the reagent.

Line 331 “M. citrata, M. spicata” should be written in italic, please check whole manuscript

Response: Thank you for your valuable comments. We have carefully checked whether the Latin names throughout the text are italicized.

Line 332, line 336 and several more have double space please check

Response: Thank you very much for your valuable advice. We have carefully reviewed the entire text and made modifications to the double space.

Above all this paper has high similarity index (around 11%) with paper https://www.sciencedirect.com/science/article/abs/pii/S0926669021006580?via%3Dihub

Which is not acceptable for high impact factor journal (or any other).

Response: Thank you for your valuable comments. The previous article was previously published by us. Because the methods in this article are based on our previous publications, resulting in high similarity. I hope that after the modification, it can be lowered.

Reviewer 3 Report

Comments and Suggestions for Authors

In this research, the authors evaluated the fumigant effect of essential oils from three species of mint, as well as their main compounds, on mortality, the activity of detoxifying enzymes, and the inhibition of acetylcholine esterase on Reticulitermes dabieshanensis. The purpose of the work was to determine the potential of essential oils and their main compounds for the control of R. dabieshanensis, as well as to provide information on the toxicity mechanisms involved. The study constitutes a relevant contribution to the development of alternative control strategies to traditional chemical synthesis insecticides.

However, it is necessary to significantly improve several aspects of the methodology, the presentation of results and expand the discussion of some topics, before the work can be published. The main aspects that need to be improved are mentioned below, and further comments are provided in the attached file.

Line 34: Which are the "traditional methods"?

Fig 1: It is suggested to eliminate these graphs, and place graphs of the probit models vs observed mortality data.

Table 2: Column X2: What statistical hypothesis does this X2 refer to? Is it to test the significance of the effect of the doses (wald test)? or is it a goodness-of-fit test? If the latter is the case, the values ​​indicate a significant lack of model fit. The methodology does not mention it and it is not clarified in the legend of the table. Although these values ​​are high, it is advisable to show the p values ​​associated with X2, to be certain about the significance or not of the test.

Table 2: Column "Regression Equation": The probit regression equation would have two parameters (intercept and slope). What is shown here looks more like the slope and its standard error. Check.

Table 2: r2: If the parameters were estimated using nonlinear regression, an r2 value should not be presented or interpreted. If the authors employed linearization (as the presentation of the r2 and f-value in the table suggests), the r2 can be interpreted in the usual way. However, the authors should clarify which estimation method they used in the statistical analysis section of the methodology

Table 2: Legend (line 110): ".... the activity of a compound was considered significantly different when the 95 % CI fail to overlap". Change instead by "...the activities of two compounds were considered significantly different when the 95% CIs fail to overlap

Line 128: When observing Fig 2 for alpha-NA, M. citrata, M. piperita, Linalool, and Carvone are in the group c, and therefore are not significantly different from control according to Duncan's test

Line 130: When observing Fig 2, for GSTs, M. citrata, and M. spicata are in the group d, and therefore are not significantly different from control according to Duncan's test

Line 137: Looking at Figure 3D, it appears that the greatest inhibition was obtained with linalool at a concentration of 6 μl/ml, which is greater than 80%. Check.

Fig 3: To convey a more objective perception at first glance, unify the Y-axis values ​​from 0 to 100 for all graphs.

Fig 3: There is almost nothing in the text about the analysis and interpretation of this figure. A broader interpretation should be given to these results.

Table 4: The same comments made in Table 2 are relevant to this table.

Table 4: Legend (line 160): ".... the activity of a compound was considered significantly different when the 95 % CI fail to overlap". Change instead by "...the activity of two compounds was considered significantly different when the 95% CIs fail to overlap"

Lines 241-242: Given that it could not be proven that essential oils or their main compounds could have an inhibitory action on detoxifying enzymes, and on the contrary, an increase in their activity was evident, the authors should discuss the possibility that R .dabieshanensis could eventually develop resistance to compounds derived from essential oils and how to prevent this risk.

Line 291: The expression you show here is incorrect. When reviewing the work of Yang et al and others above, the correct formula is E = Oa + Ob(1 − Oa). Ob is not divided by (1-Oa) but multiplied. You must correct the formula in the text, but above all, verify that the calculations of E have been carried out with the correct formula.

Section 4.8: What software or statistical language was used for the analyses?

Section 4.8: ANOVA for mortality is redundant because the best way to analyze the dose-response in terms of mortality is the LC50, which is already included. It is suggested to eliminate the ANOVA for dose-response mortality data. On the other hand, regarding the ANOVA for the inhibition data, the one-way ANOVA involves validating the assumptions of normality of the residuals and homogeneity of variances. The authors should perform these tests and mention them in this section.

Comments on the Quality of English Language

Moderate editing of the English language is necessary. 

Author Response

In this research, the authors evaluated the fumigant effect of essential oils from three species of mint, as well as their main compounds, on mortality, the activity of detoxifying enzymes, and the inhibition of acetylcholine esterase on Reticulitermes dabieshanensis. The purpose of the work was to determine the potential of essential oils and their main compounds for the control of R. dabieshanensis, as well as to provide information on the toxicity mechanisms involved. The study constitutes a relevant contribution to the development of alternative control strategies to traditional chemical synthesis insecticides.

However, it is necessary to significantly improve several aspects of the methodology, the presentation of results and expand the discussion of some topics, before the work can be published. The main aspects that need to be improved are mentioned below, and further comments are provided in the attached file.

Line 34: Which are the "traditional methods"?

Response: Thank you very much for your valuable advice. We have already “traditional methods of controlling termites often have negative impacts on 34 the environment and human health. ”was corrected as “traditional methods of chemical controlling of termites often have negative impacts on the environment and human health.”

Fig 1: It is suggested to eliminate these graphs, and place graphs of the probit models vs observed mortality data.

Response: Thank you very much for your valuable advice. We have deleted Figure 1 and placed a graph of the probability model and observed mortality data. 

Table 2: Column X2: What statistical hypothesis does this X2 refer to? Is it to test the significance of the effect of the doses (wald test)? or is it a goodness-of-fit test? If the latter is the case, the values indicate a significant lack of model fit. The methodology does not mention it and it is not clarified in the legend of the table. Although these values are high, it is advisable to show the p values associated with X2, to be certain about the significance or not of the test.

Response: Thank you very much for your valuable advice. X2 represents Pearsons chi-square, suggesting the goodness of fit test.

Table 2: Column "Regression Equation": The probit regression equation would have two parameters (intercept and slope). What is shown here looks more like the slope and its standard error. Check.

Response: Thank you for your valuable comments. We have already “probit regression” was corrected as “Slope ± SD.

Table 2: r2: If the parameters were estimated using nonlinear regression, an r2 value should not be presented or interpreted. If the authors employed linearization (as the presentation of the r2 and f-value in the table suggests), the r2 can be interpreted in the usual way. However, the authors should clarify which estimation method they used in the statistical analysis section of the methodology

Response: Thank you very much for your valuable advice. We deleted the R2 value.

Table 2: Legend (line 110): ".... the activity of a compound was considered significantly different when the 95 % CI fail to overlap". Change instead by "...the activities of two compounds were considered significantly different when the 95% CIs fail to overlap

Response: Thank you for your valuable comments. We have already a” was corrected as “two.

Line 128: When observing Fig 2 for alpha-NA, M. citrata, M. piperita, Linalool, and Carvone are in the group c, and therefore are not significantly different from control according to Duncan's test

Response: Thank you very much for your valuable advice. We added However, there were no differences in the activity of α-NA or GSTs between the M. citrata and control groups. 

Line 130: When observing Fig 2, for GSTs, M. citrata, and M. spicata are in the group d, and therefore are not significantly different from control according to Duncan's test

Response: Thank you for your valuable comments. We added However, there were no differences in the activity of α-NA or GSTs between the M. citrata and control groups. 

Line 137: Looking at Figure 3D, it appears that the greatest inhibition was obtained with linalool at a concentration of 6 μl/ml, which is greater than 80%. Check.

Response: Thank you very much for your valuable advice. We have rewritten this paragraph. The inhibitory AChE effects of Mentha spp EO against R. dabieshanensis increased significantly as the concentration increased (for M. citrata EO, F = 16.806; df = 4, 10; p < 0.001; for M. piperita EO, F =36.839; df = 4, 10; p < 0.001; for M. spicata EO, F = 46.694; df = 4, 10; p < 0.001; Fig. 3 A-C). According to Table 4, the IC50 values of M. citrata, M. piperita and M. spicata EOs against R. dabieshanensis were 18.295, 0.765 and 3.228 μl/mL, respectively. Additionally, each tested compounds (linalool and menthol) also showed a significant variation in AChE inhibitory activity at different concentrations (for linalool, F = 28.551; df = 4, 10; p < 0.001; for menthol, F = 32.160; df = 4, 10; p < 0.001; Fig. 3 D,E), but carvone has no significant difference (F = 3.433; df = 4, 10; p = 0.052; Fig. 3 F), with an IC50 of 0.136, 76.790 and 1.922 μl/mL, respectively (Table 4).

Fig 3: To convey a more objective perception at first glance, unify the Y-axis values from 0 to 100 for all graphs.

Response: Thank you very much for your valuable advice. We have unified the Y-axis values from 0 to 100 all graphs.

Fig 3: There is almost nothing in the text about the analysis and interpretation of this figure. A broader interpretation should be given to these results.

Response: Thank you for your valuable comments. We added The inhibitory AChE effects of Mentha spp EO against R. dabieshanensis increased significantly as the concentration increased (for M. citrata EO, F = 16.806; df = 4, 10; p < 0.001; for M. piperita EO, F =36.839; df = 4, 10; p < 0.001; for M. spicata EO, F = 46.694; df = 4, 10; p < 0.001; Fig. 3 A-C). According to Table 4, the IC50 values of M. citrata, M. piperita and M. spicata EOs against R. dabieshanensis were 18.295, 0.765 and 3.228 μl/mL, respectively. Additionally, each tested compounds also showed a significant variation in AChE inhibitory activity at different concentrations (for linalool, F = 28.551; df = 4, 10; p < 0.001; for menthol, F = 32.160; df = 4, 10; p < 0.001; for carvone, F = 3.433; df = 4, 10; p = 0.052; Fig. 3 E-F), with an IC50 of 0.136, 76.790 and 1.922 μl/mL, respectively (Table 4). 

Table 4: The same comments made in Table 2 are relevant to this table.

Response: Thank you very much for your valuable advice. We have also revised Table 4 according to the comments in Table 2.

Table 4: Legend (line 160): ".... the activity of a compound was considered significantly different when the 95 % CI fail to overlap". Change instead by "...the activity of two compounds was considered significantly different when the 95% CIs fail to overlap"

Response: Thank you for your valuable comments. We have already a” was corrected as “two.

Lines 241-242: Given that it could not be proven that essential oils or their main compounds could have an inhibitory action on detoxifying enzymes, and on the contrary, an increase in their activity was evident, the authors should discuss the possibility that R.dabieshanensis could eventually develop resistance to compounds derived from essential oils and how to prevent this risk.

Response: Thank you very much for your valuable advice. We have discussed this part from another perspective. We added ”Earlier studies suggested toxicity of EO’s constituents may be related to the cell damage in insect tissues [48-50]. Given the importance of these enzymes in cell protection from plant allelochemicals, the induction of its activity can be considered as a defense mechanism in R. dabieshanensis against EOs.”

Line 291: The expression you show here is incorrect. When reviewing the work of Yang et al and others above, the correct formula is E = Oa + Ob(1 − Oa). Ob is not divided by (1-Oa) but multiplied. You must correct the formula in the text, but above all, verify that the calculations of E have been carried out with the correct formula.

Response: Thank you for your valuable comments. Very sorry, we mistakenly multiplied Ob by (1-Oa) and wrote it as Ob divided by (1-Oa).

Section 4.8: What software or statistical language was used for the analyses?

Response: Thank you for your valuable comments. We added All the analyses were done using SPSS (version 19.0; SPSS Inc., Chicago, IL, USA).

Section 4.8: ANOVA for mortality is redundant because the best way to analyze the dose-response in terms of mortality is the LC50, which is already included. It is suggested to eliminate the ANOVA for dose-response mortality data. On the other hand, regarding the ANOVA for the inhibition data, the one-way ANOVA involves validating the assumptions of normality of the residuals and homogeneity of variances. The authors should perform these tests and mention them in this section.

Response: Thank you very much for your valuable advice. We have rewritten this paragraph. The results were expressed as means ± standard errors. The doses causing 50% and 90% (LC50 and LC90 respectively) mortality were determined by probit analysis using Online Tool (OPSTAT) (http://14.139.232.166/Probit/probitanalysis.html). The inhibition rate were tested through the analysis of variance (ANOVA) followed by Duncan’s new multiple range method Significant Difference test at P = 0.05 level of significance using SPSS (version 19.0; SPSS Inc., Chicago, IL, USA).

Round 2

Reviewer 2 Report

Comments and Suggestions for Authors

My main concern about this study remains unanswered and it is about GC MS analysis. Presented chromatograms are cut on 22 min, and we can not see are there some compounds later. Essential oils are generally complex mixtures of compounds. And in this study are only 10 or less compounds detected which led me to the conclusion that the applied method is not appropriate. Please discuss how this method was developed, using some literature data or is it a commercial one? How did you decide to use exactly this method?

Regarded to added reference about the rapid degradation of plant essentials (24th reference), I carefully read this paper and I couldn’t find the mentioned discussion. As I previously stated, some compounds from essential oil can be easily volatilized but that is far away from “rapid degradation”. Please use more appropriate terms.

Author Response

My main concern about this study remains unanswered and it is about GC MS analysis. Presented chromatograms are cut on 22 min, and we can not see are there some compounds later. Essential oils are generally complex mixtures of compounds. And in this study are only 10 or less compounds detected which led me to the conclusion that the applied method is not appropriate. Please discuss how this method was developed, using some literature data or is it a commercial one? How did you decide to use exactly this method?

Response: Thank you for your valuable comments. I'm very sorry to have caused you such doubts. Our GC-MS heating program is based on the previous method. Previously, we also conducted a pre-experiment with a set time of 30 minutes, and after 15 minutes, there was basically no peak. In addition, we also measured some other essential oils, and the analysis results showed many components. I think the only reason that can be explained may be that all three essential oils used in this experiment were commercially available, rather than extracted by oneself. We have introduced it in the materials section.

Regarded to added reference about the rapid degradation of plant essentials (24th reference), I carefully read this paper and I couldn’t find the mentioned discussion. As I previously stated, some compounds from essential oil can be easily volatilized but that is far away from “rapid degradation”. Please use more appropriate terms.

Response: Thank you very much for your valuable advice. We have already “rapid degradation” was corrected as “easy to volatile ”.

Reviewer 3 Report

Comments and Suggestions for Authors

The authors considered most of the suggestions to improve the writing substantially. However, there are still a few aspects that are important to review before publishing the work:

Tables 2 and 4: X2. If this Chi-square value corresponds to a test of goodness of fit of the model with respect to the observed data, as the authors indicate, it would be expected that it would not reject the null hypothesis if the fit of the model to the data is adequate. In this case, these values ​​suggest that the null hypothesis is rejected, so the models would not present a good fit to the observed data. However, it cannot be known conclusively because there are no p-values ​​associated with the test. Authors should review this carefully.

Lines 98 and 149: Change "compound" to "compounds"

Table 3: This chi-square does not inform whether the null hypothesis is accepted or rejected, because no associated p-value is presented.

Authors should have the text reviewed by an English native speaker

The comments are also included in the attached file. 

Comments on the Quality of English Language

Authors should have the text reviewed by an English native speaker

Author Response

The authors considered most of the suggestions to improve the writing substantially. However, there are still a few aspects that are important to review before publishing the work:

Tables 2 and 4: X2. If this Chi-square value corresponds to a test of goodness of fit of the model with respect to the observed data, as the authors indicate, it would be expected that it would not reject the null hypothesis if the fit of the model to the data is adequate. In this case, these values suggest that the null hypothesis is rejected, so the models would not present a good fit to the observed data. However, it cannot be known conclusively because there are no p-values associated with the test. Authors should review this carefully.

Response: Thank you for your valuable comments. χ2 value, Pearsons chi-square, suggesting the goodness of fit test, not significant (ns) at p > 0.05.

Lines 98 and 149: Change "compound" to "compounds"

Response: Thank you very much for your valuable advice. We have already “compound” was corrected as “compounds”.

Table 3: This chi-square does not inform whether the null hypothesis is accepted or rejected, because no associated p-value is presented.

Response: Thank you for your valuable comments. X2 in Table 3 does not represent chi square, but rather represents the antagonistic, additive, or synergistic effects of binary mixtures. χ2 = (Om-E)2/E

Authors should have the text reviewed by an English native speaker

Response: Thank you very much for your valuable advice. We have had native English speakers review the text.